# Urban Heat Mitigation towards Climate Change Adaptation: An Eco-Sustainable Design Strategy to Improve Environmental Performance under Rapid Urbanization

**Mehdi Makvandi [1,2,3]**, **Wenjing Li [1]**, **Xiongquan Ou [1]**, **Hua Chai [1]**, **Zeinab Khodabakhshi [2]**, **Jiayan Fu [1]**, **Philip F. Yuan [1,\*]** and **Elyse de la Joie Horimbere [3]**

1   College of Architecture and Urban Planning, Tongji University, Shanghai 200092, China
2   College of Civil Engineering and Architecture, Wuhan University of Technology, Wuhan 430070, China
3   College of Architecture and Urban Planning, Huazhong University of Science and Technology, Wuhan 430070, China
\*   Correspondence: philipyuan007@tongji.edu.cn; Tel.: +86-139-0173-5092

**Abstract:** Rapid urbanization has led to drastic land-use/cover changes (LUCCs) and urban heat islands (UHIs), negatively altering the urban climate and air quality. LUCC's significant impacts on human health and energy consumption have inspired researchers to develop nature-based solutions to mitigate UHIs and improve air quality. However, integrating GIS-CFD modeling for urban heat mitigation towards climate change adaptation was largely neglected for eco-sustainable urban design in rapidly urbanizing areas. In this study, (1) long-term LUCC and meteorological analysis were conducted in the Wuhan metropolitan area from 1980 to 2016; (2) to mitigate the adverse effects of LUCC under a speedy development process, the role and relevance of optimizing building morphology and urban block configuration were discussed; (3) and particular design attention in strategy towards climate change adaptation for environmental performance improvement was paid in Wuhan's fast-growing zones. The results show that UHII in 1980 was less severe than in 2016. Air temperature (Ta) increased by 0.4 °C on average per decade in developing areas. This increases the severity of UHII in urban fringes. It is found obligatory for a nature-based design to adopt urban morphology indicators (UMIs) such as average building height ($\mu$BH), sky view factors ($\psi_{SVF}$), and building density (BD/$\lambda_P$ = % of built area) towards these changes. Further, on-site measurement revealed that $\lambda_P$ is the most effective indicator for increasing urban heat around the buildings and boosting UHII. Using UMIs and a combined three-in-one regulation strategy based on $\mu$BH of common building types of high-rise (BH[A]), mid-rise (BH[B]), and low-rise (BH[C]) buildings can effectively contribute to regulating Ta and air movement within block configuration. As a result of this study's strategy, urban heat is mitigated via reinforcing wind in order to adapt to climate change, which impacts the quality of life directly in developing areas.

**Keywords:** urbanization; land-use/cover changes (LUCCs); urban heat island intensification (UHII) urban morphology indicators (UMIs); ventilation performance strategies (VPSs)

## 1. Introduction

Urban heat islands (UHIs) are mostly caused by urbanization and climate change [1], which refer to the phenomenon that urban air temperature due to the thermal properties of the urban tissue is considerably higher than its surroundings and rural areas [2]. It has been found that urbanization processes at a global scale have largely been focused in China during recent years [3]. With an increase in rapid urbanization without the right planification (false or pseudo-urbanization processes [4]) and accelerating the process of land-use/cover changes (LUCCs), the UHI phenomenon has become increasingly more serious [5]. Undesired effects of UHIs generate many grave problems closely related to humans and the environment: deteriorating air quality [6], threatening human comfort and

well-being [7], increasing energy and water consumption [8,9], etc. According to the United Nations' Global Urbanization Trend (UN Forecast, 2018) report [10], global urbanization rose from 24% in 1950 to 55% in 2018. As an inevitable process of human development, this trend is expected to reach 68% by 2050 [11–15]. Besides the increase in UHI intensity (UHII) and land surface temperature (LST) [16], major pollutants in urban air such as carbon monoxide (CO), fine particles ($PM_{2.5}$), coarse particles ($PM_{10}$), nitrogen dioxide ($NO_2$), sulfur dioxide ($SO_2$), and ozone ($O_3$) are also on the rise [17]. Hence, it is urgently vital to understand how to adapt to climate change and mitigate UHII, making it a research hotspot under an urgent demand for urban ecology and urban climatology [18,19].

In the literature and practice, various outdoor space strategies have been proposed to mitigate UHII, among which urban green space growth is one of the most common and popular strategies [20–22]. Green spaces with shading can reduce the proportion of impervious surfaces directly exposed to solar radiation, and evapotranspiration can remove heat from the air [23]. Furthermore, urban blue spaces such as ponds [24], wetlands [25], lakes, and rivers [26] have also demonstrated a remarkable cooling potential within their surrounding areas due to the high evapotranspiration rates and capacity to absorb heat. Apart from these naturally derived UHII mitigation and control measures, urban geometry [27], building density ($BD/\lambda_p$ = % of built area) [28], aspect ratio (building height/street width (H/W)) [29], built environment albedo increment (e.g., reflective pavement) [18], layout, form, and fabric [30–32] would also affect local climate and thermal comfort. Meanwhile, the relationship between built heights and street widths (street metrics) is highly important to estimate UHI and formulate urban typologies [33]. These findings led to further studies indicating that the ventilation performance of blocks is greatly influenced by their configuration [34,35], which further affects the dispersion of air pollutants and UHII.

In order to link urban spatial configuration to UHII, different urban morphology indicators (UMIs), such as average building height ($\mu BH$), sky view factors ($\psi_{SVF}$), etc., have been developed and used [36,37]. Following the relationship outlined above, corresponding guidelines and recommendations for UHII mitigation were presented to decision-makers for developing cities that are climate-adaptive. Particularly, many related studies in the field of remote sensing have been conducted by exploring the relationship between LST and vegetation coverage ratio/spatial pattern, fabricating 2D/3D morphology characteristics [38–40]. This can be combined with aerial images and LiDAR to track LUCC and obtain 3D information (such as building height, which is a fundamental factor in UHI) [41]. However, a major flaw lies in the fact that LST obtained from remote sensing is not a direct parameter signifying thermal sensation. Therefore, remote sensing studies cannot establish a direct relationship between human thermal comfort and urban morphology. Air quality is linked to human comfort [42], and a low quality may result in lower pollutant distributions and higher UHIIs in actual urban blocks, which makes their relationship to block configurations crucial. Further research is needed to clarify and grasp this relationship due to complicated morphologies in complex urban environments.

Several studies have combined measurements on-site with information collected via remote sensing [43,44]: climatic parameter sensors were placed in the study area, and remote sensing technology collected morphological data surrounding the measuring points. The combination of these two measures can provide a feasible method for urban morphology and thermal comfort level (i.e., air temperature (Ta) and relative humidity (RH)) to be directly related. However, this method's reliability and precision depend on where the measuring points are placed. A study area's thermal conditions may not be accurately represented by meteorological parameters such as Ta obtained from one or more measurement sites. Luckily, advancements in computer technology and numerical calculation have made it possible to conduct reliable studies and simulation experiments in order to understand more thoroughly the correlation between urban morphology and local thermal comfort. In addition to providing detailed information on the thermal comfort of the study area, the computer simulation model can also assess the relationship between

urban morphology and thermal comfort in built-up areas [45,46]. In urban planning and design, the most influential urban form and layout can also be explored [47,48].

As part of this study, Wuhan's constantly changing morphology under rapid development and accelerated urbanization that significantly affects wind and thermal environments is considered. Land-use classification (Table A1) was determined and evaluated to highlight major structural and ecological changes. Analysis of the LUCC was obtained by using ArcMap as a main component of Esri's Arc geographic information system (GIS) [49].

This work mainly consists of two parts (Figure 1). In part one, the LUCC process was tracked to find out the changes in urban form and environmental conditions under rapid urbanization through high-resolution remote sensing images (GIS and remote sensing-based approach) [48] and meteorological long-term data analysis. Morphological changes were clustered and screened as influential parameters for UHII and air movement in the most transformed urban areas, particularly residential blocks. The morphological parameters and forms of typical representative residential blocks were selected. In part two, the computational fluid dynamics (CFD) [50–52] simulation technique of the selected representative blocks was conducted, and the ventilation performance and UMIs to enhance air movement and thermal comfort were analyzed, which can mitigate UHII and pollutant concentration. In this regard, the impact of the various block types on the wind environment was analyzed and investigated. Finally, the differences between residential block typologies were compared based on ventilation performance assessment in accordance with morphological indicators. Results from this study are expected to improve our understanding of the influences of urban spatial configuration on the wind environment under rapid urbanization and provide valuable suggestions to urban planners about how to mitigate UHII and build a thermally comfortable outdoor space in urban settings.

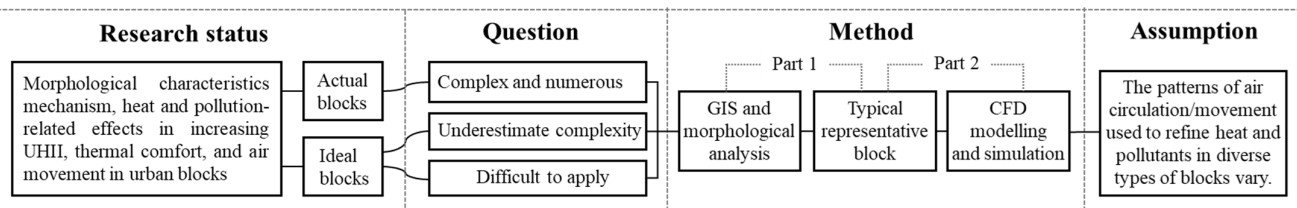

**Figure 1.** Research logic diagrams.

The remainder of the paper is structured as follows: Section 2 presents the methodologies used in this study, describing the simulation model, scenarios setting, and data analysis method; Section 3 presents the study results and discusses the analysis of the main study results; in Section 4, conclusions are drawn based on the results and analysis.

## 2. Materials and Methods

Four distinct aspects and approaches were applied to develop targeted urban design strategies based on the concept of local and telecoupling coordination models of urbanization, eco-environment [53], and their interrelationships with LUCC (Figure 2). These aspects require a careful evaluation to regulate urban microclimates that are correlated to eco-environment systems (EESs) and mitigate UHII effects during rapid urbanization. These four aspects and corresponding scenarios are further discussed in Sections 2.2–2.5.

### 2.1. Study Area

The city of Wuhan (south central China) lies along the middle reaches of the Yangtze river at the confluence of the Han and Yangtze rivers [53]. The development of Wuhan has been accelerating, especially after its economic reform (Figure 3d), ranking it among the top fastest-growing cities. Figure 3d reveals the history of Wuhan's urbanization and its upward linear trend over the stage of the city's development. Spatial patterns and hotspots of China's annual mean air quality index (AQI) indicate Wuhan as one of the most concerning hotspot zones with 90% confidence [54], highlighted in Figure 3a. Wuhan

occupies a land area of 8573 km$^2$, 888 km$^2$ of which is the built-up area. By 2013, Wuhan's population had reached 10.22 million, and 795 km$^2$ of land was used for urban construction in the built-up area. The population of the built-up area had grown to 4.4 million. Wuhan's summer period lasts almost 135 days a year, and the city has a historical reputation for its harsh local climate. During its rapid urbanization, Wuhan has undergone a severe UHII effect, particularly in developing areas, as shown in Figure 3c. For the simulation study, we performed land-use analysis in various zones and examined urban form more closely in the most remarkably transformed zones.

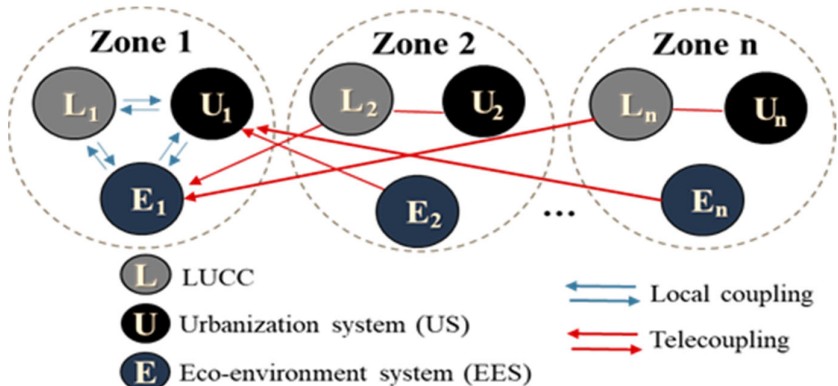

**Figure 2.** The schematic diagram for local and telecoupling of US, EES, and LUCC.

*2.2. GIS and LUCC Long-Term Observation along with Meteorological Data Analysis (1980–2016)*

In the first step, long-term observation [55] of meteorological data [56,57], climate factors [58], and remote sensing images [59] during the period 1980–2016 were applied and analyzed. For this purpose, GIS data and digital elevation models (DEMs) with raster maps of Wuhan City (8573 km$^2$) at ten-year intervals from 1980 to 2016 were established and carefully analyzed using specified raster information. Afterward, satellite remote sensing techniques (SRSTs) coupled with GIS [60] approaches were used, and data were analyzed by ArcMap 10.4.1 with the property of cell size (X, Y) and value 30, 30 and a spatial reference system of one meter. Accordingly, land-use changes were carefully assessed and measured due to the long-term impact of rapid urbanization. In 2016, the urban land cover area was 795 km$^2$, a 3.21 times increase from 220.5 km$^2$ (1980) to about 489.3 km$^2$ (2016), and the water surfaces (WSs) [61] declined by 145.64 km$^2$ (16% of the total range) in the urban area of Wuhan. As predicted [62], the rate of urbanism may reach 84% and the population of the built-up spaces will be 5.03 million, where the total population will increase by about 11.8 million. If the growth of the city continues this trend, by 2025 the decrease is predicted to be about 204.45 km$^2$ (22%). At this point, it is purposed to determine the long-term impact of the urbanization process on climate factors (e.g., air temperature ($T_a$); relative humidity (RH)) to obtain the accrued data which are needed to find the significant changes that have been linked with the greatest climate and land-use changes [63] over time during urban transformation.

*2.3. Comparative Analysis of Developing Areas and Site Selection*

As the second step, data collection for the long-term observational study (1980–2016) was conducted from the weather stations at Caidian (A,30°34′54.5″ N 114°01′45.6″ E) Huangpi (B,30°52′55.7″ N 114°22′32.4″ E), Xinzhou (C,30°50′26.2″ N 114°48′03.2″ E), and Jiangxia (D,30°22′29.8″ N 114°19′15.8″ E). This information was collected from the metrological bureau of Wuhan. Due to the long-term observation, tipping points were selected to represent the most significant changes regarding the transformation of water surfaces into urban construction land. Accordingly, and due to the major objective of this study, quantitative evaluation and measurement of these changes in the urban microclimate were undertaken in different residential urban blocks under the impact of UHII. These are

classified as BH$^A$, BH$^B$, and BH$^C$; high-rise buildings (10~34 stories), mid-rise buildings (7~9 stories), and low-rise buildings (1~3 stories), respectively [64]. All the changes discovered in land-use transformation during Wuhan City's urbanization from 1980 to 2016 can be seen in Figure 3d, where urban land increases (red color) and water surface declines (blue color). These urban morphological changes seriously deteriorated air quality and increased UHII, notably where pervious surfaces are replaced with impervious surfaces in developing areas. This led to an increase in urban heat and temperatures, significantly in newly developed residential areas. Therefore, strategies for mitigating and adapting to urban heat are crucial in developing areas.

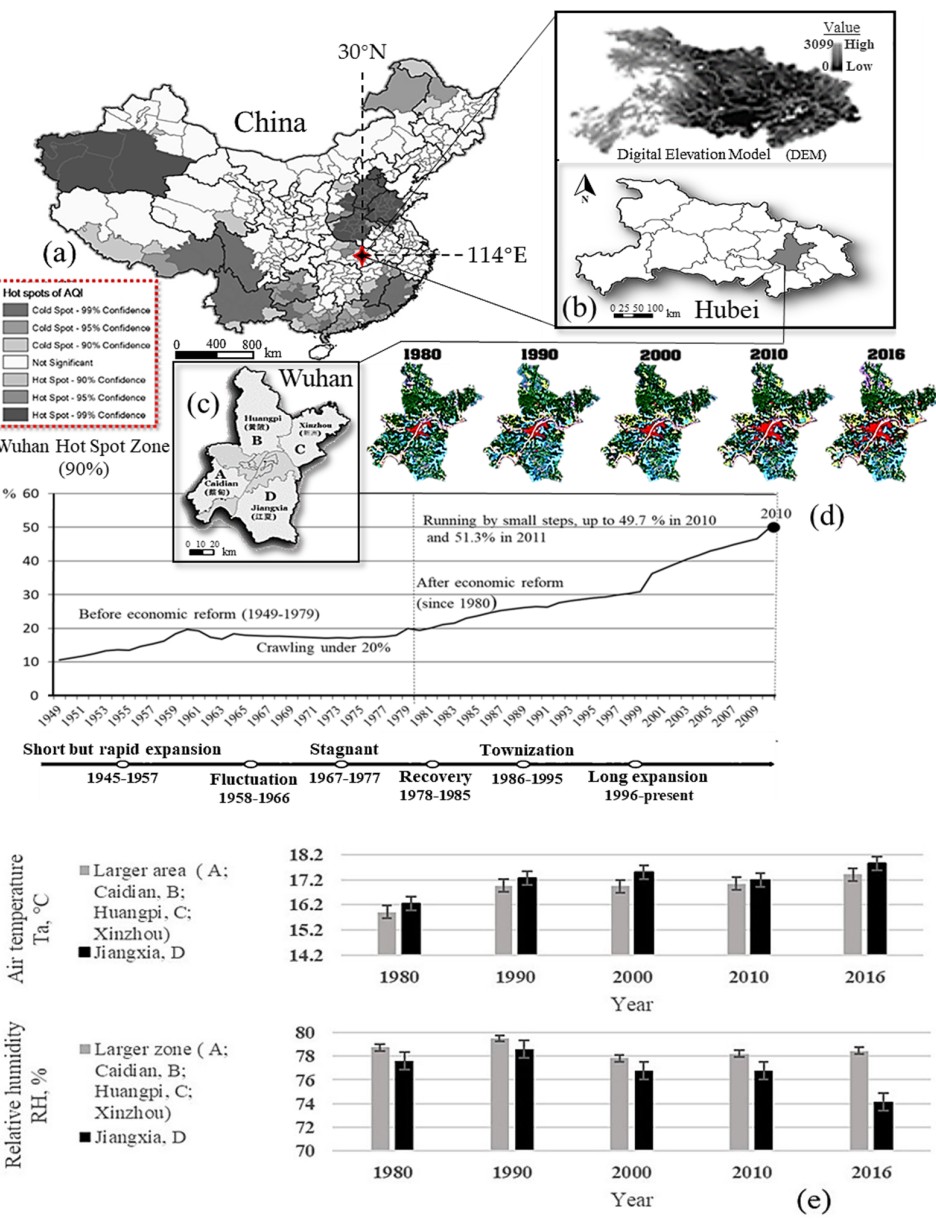

**Figure 3.** Urbanization system (US) effect of Wuhan on LUCC and regional climate change within EES. (**a**) China's hot stops of AQI specified Wuhan as one of the major hotspot zone (90%), (**b**) the digital evaluation model (DEM) of Hubei province and the area of Wuhan, (**c**) Wuhan developing areas, (**d**) long-term urbanization analysis with emphasis on study duration after economic reform since 1980 in Wuhan (red represents the urban area that rapidly increased from 1980 to 2016, and blue represents the water body that decreases at the same time and deteriorated air quality), (**e**) Wuhan's meteorological long-term analysis (1980–2016).

### 2.4. On-Site Fixed and Mobile Observation and Measurements

The third step involves on-site measurement using mobile observation [65] of meteorological data, remote sensing images, and climate factors focusing on air temperature (Ta), relative humidity (RH), solar radiation, and wind environment. Therefore, in order to study the influence of block morphology (form and height) on urban microclimates, fixed and mobile observation was adopted [66]. The main equipment and instruments include a small weather station involving wind speed and direction indicator, an anemometer (placed 1.5 m above ground and using JTSOFT Meter V1.3 software to extract the data), a small portable temperature and humidity meter (data-logger using TRLog software to extract the data), mobile camcorder, and a handheld global positioning system (HhGPS). For the mobile observation approach, a walking method at a speed of 3 km/s for all blocks was used in the morning (5:30–6:30 am), at noon (13:00–14:00), and at night (21:00–22:00). Observations and measurements were recorded simultaneously for all blocks within the area of 1 km.

### 2.5. Numerical CFD Simulation Models Adaptable to Urban Growth

The fourth step is a simulation method using CFD analysis through the standard K-ε model for the turbulent model in the FLUENT solver [67–70]. This numerical simulation was applied to create models and simulate the objects in order to tackle the numerical problems in which digital models of actual residential blocks are generated (using AutoCAD 2018). The CFD domain [71] was set up after adding proper mesh [72] (staggered mesh [69]) on models by ICEM-CFD 15. The study performs to evaluate urban microclimates with the interactive impact of building diversity on the thermal balance and urban heat that involves wind flows [71,72]. Figure 4 illustrates the changing phase of urbanization (urban form, water body, and air movement) and analyzes an actual case from reality to virtuality model after a long-term regional study. Appendix B shows the governing equations as well as model constants validated and used in the numerical analysis [70,71].

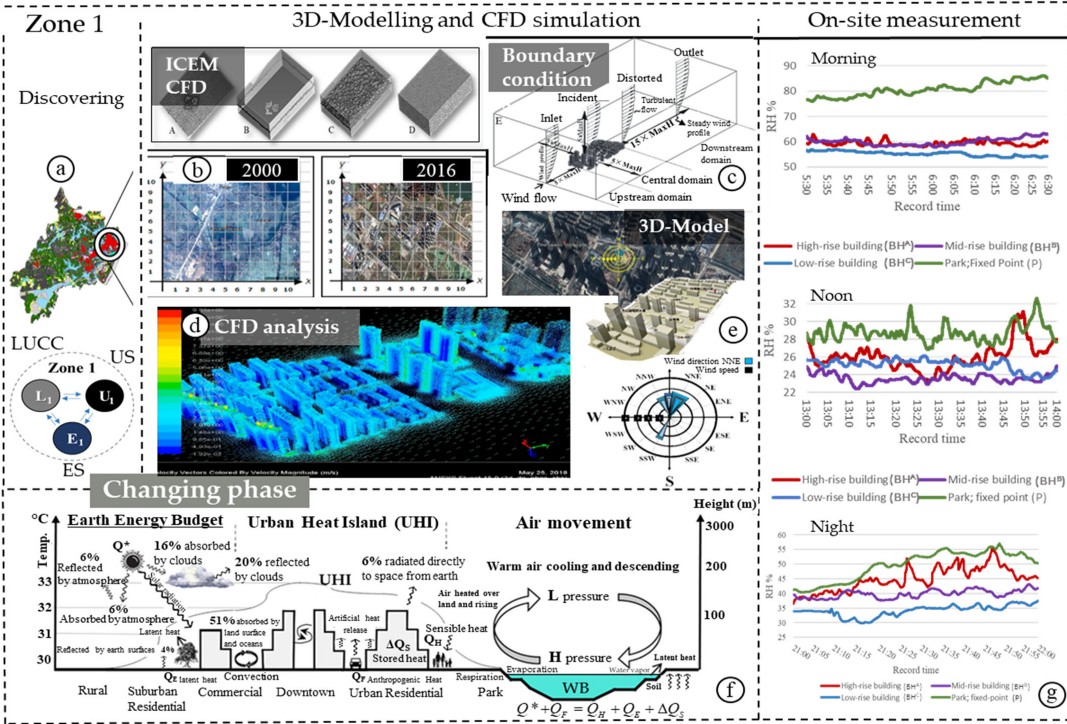

**Figure 4.** Analyzing an actual case from reality to virtuality model after a long-term regional study. (**a**) Caidian district; (**b**) plane coordinate system from 2000 to 2016; (**c**) ICEM CFD grid mesh with boundary conditions (CFD domain); (**d**) computational fluid dynamic (CFD) analysis; (**e**) actual 3D model generated for the study and the mean average of prevailing wind; (**f**) changing phase of urbanization and LUCC; and (**g**) experimental study.

*2.6. Typological Strategy of Block Form Design to Reinforce Wind and Mitigate UHII*

A typological design strategy evaluation was conducted to assess the impact of building morphology on the wind environment and UHII mitigation. First, highlighted points of significant changes were simulated to show the effect of new morphology on air movement. Secondly, building typology's impact on the increasing air movement in city blocks was considered. In this context, a number of typologies comparable in characteristics and living space layouts were simulated using a Fluent solver with the steady-state Reynolds-averaged Navier–Stokes (RANS) equations and CFD post-version 15 code. The typologies illustrate actual city growths constructed during urban formation and transformation in Wuhan, China. The building's effect on the outdoor wind environment to reinforce the air around the buildings is a goal that can efficiently relieve the heat stress accumulated between building surfaces and the negative effects of the LUCC. This strategy also can be considered for air pollutant removal and providing fresh air for inhabitants. Configuration, layout, compactness, direction, and forms are investigated in this regard.

## 3. Results and Discussion

### 3.1. LUCC and Climatic Changes in Wuhan: A Comprehensive Long-Term Study

To achieve urban design strategies, understanding chronological land-use and climate changes are essential for future sustainable urban design. Hence, a wide scale of land-use transformation was analyzed under the rapid urbanization of Wuhan (Figure 5).

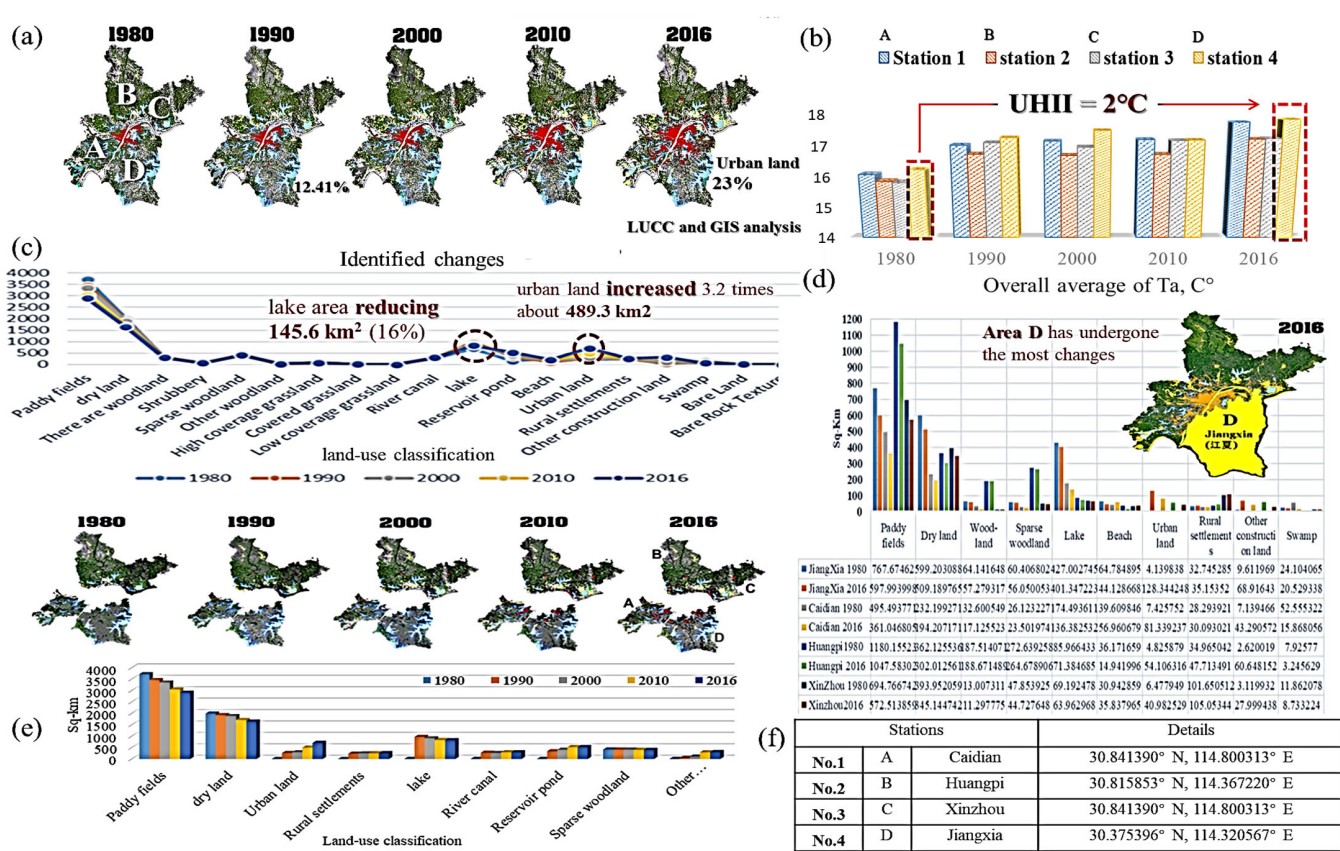

**Figure 5.** An analysis of Wuhan's land-use transformation under rapid urbanization. (**a**) LUCC and GIS analysis; (**b**) UHII in developing areas of A, B, C, and D; (**c**) identified changes due to Wuhan's land-use classification; (**d**) clarifying the most affected area; (**e**) a comparative analysis between area D as the most affected area by the changes with the other developed regions in A, B, and C; (**f**) geographic coordination (latitude and longitude) of the stations.

A long-term study of LUCC using the land-use classification of Wuhan (Figure 5c) reveals that urban morphology is highly affected by rapid development, with significant changes in developing areas (Table 1). Increasing urbanization has resulted in a reduction in blue spaces in Wuhan since 1980. Meanwhile, paddy fields, dry-lands, woodlands, shrubbery, sparse woodland, high-coverage grassland, covered grassland, low-coverage grassland, beach, swamp, bare lands, and bare rock texture decreased, and subsequently, urban lands increased by 1173.6238% in total. In 1990, land-use transformation accompanied by the water surface reduction (145.64 km$^2$, 16%) became more severe, and urban lands dramatically increased (489.3 km$^2$) in the urban fringe. As a result of LUCC, water surfaces have been transformed into urban lands, affecting thermal balance and reducing natural ventilation significantly. These eco-environmental changes significantly affected the Jiang Xia and Caidian climates. Climate changes in Jiangxia region were most pronounced with higher relative humidity (4.05%) and temperature (0.7%) (Figure 3e). Region-to-region meteorological observations indicate that UHII has increased by 2 °C in newly built-up areas (Figure 5b). Figure 5f shows the latitude and longitude of the weather stations. Therefore, reinforcing the air movement is vital to mitigate urban heat in Wuhan megacity with rapid development. To end, long-term observation indicates that the UHII in 1980 was less severe than in 2016, significantly before water surfaces were converted into urban lands. Ta increased by 0.4 °C on average per decade. It is also evident from the reduced water area, that nighttime ventilation is not as effective as it once was.

**Table 1.** Discovering LUCC based on land-use classification of Wuhan, Appendix A.

| No | LUCC | Value % | Decline | Increase | No | LUCC | Value % | Decline | Increase |
|---|---|---|---|---|---|---|---|---|---|
| 1 | Paddy fields | 1.279965% | ☑ ↓ | ☒ | 8 | $^{LC}$ Grassland | 1.56226% | ☑ ↓ | ☒ |
| 2 | Dry-lands | 1.214954% | ☑ ↓ | ☒ | 9 | Beach | 1.147242% | ☑ ↓ | ☒ |
| 3 | Woodlands | 1.087316% | ☑ ↓ | ☒ | 10 | Swamp | 2.35544% | ☑ ↓ | ☒ |
| 4 | Shrubbery | 1.019609% | ☑ ↓ | ☒ | 11 | Bare lands | 1.517368% | ☑ ↓ | ☒ |
| 5 | Sparse wood | 1.047605% | ☑ ↓ | ☒ | 12 | Bare rock $^{texture}$ | 1.616842% | ☑ ↓ | ☒ |
| 6 | $^{HC}$ Grassland | 1.017211% | ☑ ↓ | ☒ | 13 | **Urban lands** | **1173.6238%** | ☒ | ☑ ↑ |
| 7 | $^{C}$ Grassland | 1.175336% | ☑ ↓ | ☒ | | | | | |

3.1.1. Regional LUCC Comparison among Fastest-Growing Areas

Further LUCC analysis in fast-growing regions indicates that the most decline in farmlands (paddy fields) has occurred in Caidian (27%), Jiangxia (22.1%), Xinzhou (17.59%), and Hunagpi (11.23%). Dry-lands have declined the most in Hunagpi (16.6%), Caidian (16.36%), Jiangxia (15.02%), and Xinzhou (12.38%). Sparse woodlands have declined most in Caidian (10.03%), Jiangxia (7.21%), Xinzhou (6.53%), and Hunagpi (2.91%). Caidian (69.8%), Hunagpi (59%), Xinzhou (26.37%), and Jiangxia (14.83%) have experienced the greatest declines in the swamp. The most significant declines in lakes have occurred in Jiangxia (approximately 1%), Xinzhou (0.92%), Hunagpi (0.83%), and Caidian (0.78%). In urban fringes, however, lakes have declined dramatically since the 1990s. Figure 5d illustrates the most affected growing regions. A comparative analysis of area D (Jiangxia) as the most affected area by the changes with the other developed regions in A (Caidian), B (Hunagpi), and C (Xinzhou) is revealed in Figure 5e. Meteorological long-term analysis of Ta and RH (1980–2016) specifies that Jiangxia area was one of the most significant regions affected by LUCC under the rapid urbanization of Wuhan. Figure 3e compares Jiangxia (D) to the above fast-growing regions (A, B, and C) in an urban environment.

### 3.2. Experimental and Simulation Study

#### 3.2.1. The Impact of Urban Block Morphology on Urban Microclimates

Figure 6a illustrates selected areas in the D zone where the most significant LUCC has occurred. The case setting (average value) and measurement strategies are shown in Figure 6b. Figure 6c depicts the daytime air temperatures of urban residential sampling areas (building types of BH$^A$, BH$^B$, and BH$^C$). These cases illustrate how common building types and morphologies affect Ta and RH's differences over time.

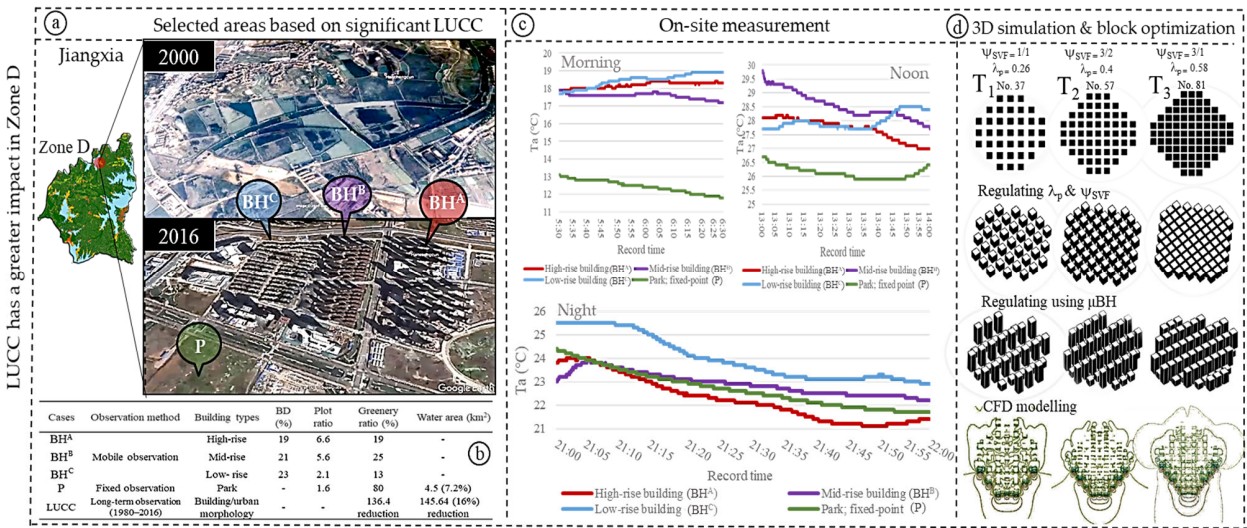

**Figure 6.** Experimental and simulation study. (**a**) Selected areas in zone D based on significant LUCC; (**b**) case setting (average value); (**c**) on-site measurement in the morning, at noon, and night using fixed and mobile observation within residential blocks; (**d**) 3D simulation and block optimization.

The results revealed a significant difference of 5.4 °C between case P (greenery ratio/GR= 80% and water body/WB = 7.2%) and other built-up environments in urban residential blocks in the early morning. This area is the coldest area with a significant air temperature (Ta) difference on the block atmosphere, as obtained from the comparative Analysis in the morning. Simultaneously, the Ta in case BH$^C$ is higher than in other cases because BD/$\lambda_p$ = 23% is higher than other types. Thus, a link between Ta change and BD/$\lambda_p$ ratio can be well-defined. A comparison of the cases' plot ratio (PR) also revealed the importance of this ratio in heat balance and Ta change. In addition to BD and PR, a combined three-in-one regulation strategy based on µBH of analyzed common building types (BH$^A$_BD = 19%, BH$^B$_BD = 21%, and BH$^C$_BD = 23%) can effectively contribute to regulating Ta and air movement within block configuration. Further analysis at night confirms that Ta distribution varies based on block morphology indicators, with P having a higher distribution than a BH$^A$ and a lower distribution than BH$^C$ and BH$^B$. At noon, according to the reflection, transmission, and absorption of heat gained all over the land surfaces, the Ta in BH$^A$ and BH$^B$ declined, except in case BH$^C$, which increased because of the higher BD ratio, lower GR, and PR. Accordingly, gained heat is accumulated due to the reduction in air movement in a densely built-up area which changes the heat balance and increasingly affects the Ta, which rises by about 1 °C. This difference is even higher between BH$^C$ and P area by 2 °C. We also found that BH$^A$ with a high-value PR can increase the air movement around the building and force the wind at the top of the buildings downwards to the earth. Air change rates in BH$^A$, BH$^B$, and BH$^C$ were 3 m/s, 2.4 m/s, and 1.2 m/s, respectively.

Lastly, a substantial impact of BD and PR as morphological indicators and GR and WB on the surrounding environment was evident, which can remarkably change the heat balance and urban microclimate. These changes show the significant effect of UMIs/city block morphology on the built-up microclimate, which surface urban heat island/SUHI

intensifies in developed areas. Considering the natural insulating properties of air, it is difficult for heat to be transferred from one place to another. This is even more apparent in urban blocks with higher BDs where there is a shortage of air movement and surfaces affected by the sun's heat gradually heat the air layers around the building surfaces. Due to $BH^C$'s density, accumulated heat and Ta enlargement were more evident, especially in the morning and night (Figure 6b). Outcomes reveal that the gained heat between urban blocks gradually transferred to the air layers near the earth but not to the higher air layers, thereby impacting the urban microclimate. Hence, block optimization and regulating $\lambda_p$ (BD), $\psi_{SVF}$, and $\mu BH$ are implemented using advanced computer-based technology (Figure 6d) to find nature-based sustainable strategies to enhance air circulation for heat removal around the building and improve environmental performance (Section 3.2.2).

### 3.2.2. UHII Mitigation Strategies Based on Building Layouts and Typologies

An evaluation highlighted here explores the effectiveness of design strategies based on building layouts and typologies to improve wind environments and mitigate UHII. The eco-sustainable design strategy was the focus of this study. First, highlighted points with great LUCC were simulated to show the effect of new morphology on air movement (Figure 7a). Second, the impact of building typology to increase air movement in urban blocks was considered. Hence, different typologies with comparable characteristics and layouts were simulated using a Fluent solver with the steady-state Reynolds-averaged Navier–Stokes (RANS) equations and CFD post-version 15 code. The typologies are illustrative of real city growths constructed during urban formation and transformation in Wuhan (Figure 7e). The building effect on the outdoor wind environment to reinforce the air around the buildings is a goal that can relieve the heat stress accumulated between building surfaces and the negative effects of UHIs efficiently. This strategy also can be considered for air pollutant removal and providing fresh air for inhabitants. In this regard, configuration, layout, compactness, direction, and forms concerning $\lambda_p$, $\psi_{SVF}$, and $\mu BH$ are investigated and an ideal block is proposed (Figure 7g).

### 3.2.2.1. Wind Environment Analysis around Buildings within Urban Blocks

As the second region with the largest transformation of farmlands and water bodies into built-up areas, Caidian has been considered for practicing new cases after Jiangxia. There was a dramatic drop in air movement around the buildings in these areas as a result of the morphological changes, so a simulation technique was implemented to assess the current changes in recently formed blocks (Figure 7a). To investigate further, a turbulent and incompressible fluid flow model around the buildings is considered and the continuity and RANS equations [30,73] were performed around the buildings to provide a solution. Hence, we investigated a typological nature-based sustainable strategy following an integrated analysis and proposed an optimal design for enhancing air movement and mitigating UHII through eco-efficient practices (Figure 7g). As shown in Figure 4d, CFD Modeling was applied. The simulated model revealed that wind flow is weakened by incremental surface roughness (0.7–1.5) due to building mass or blocked by large obstructions in the urban canopy layer which by experimental cases also validated. Analyzing all modeled actual cases within urban blocks shows that building arrangements or block configurations affecting wind environments lack appropriate design strategies. Thus, all modeled blocks led to a decline in air circulation, indicating that adjusting the blocks is a nature-based solution to reinforce air movement in densely built-up areas experiencing rapid development. As can be seen in Figure 7a, morphological indicators such as $BD/\lambda_p$ and $\psi_{SVF}$ consideration can affect the wind environment. This requires further investigation, which is described in the following section.

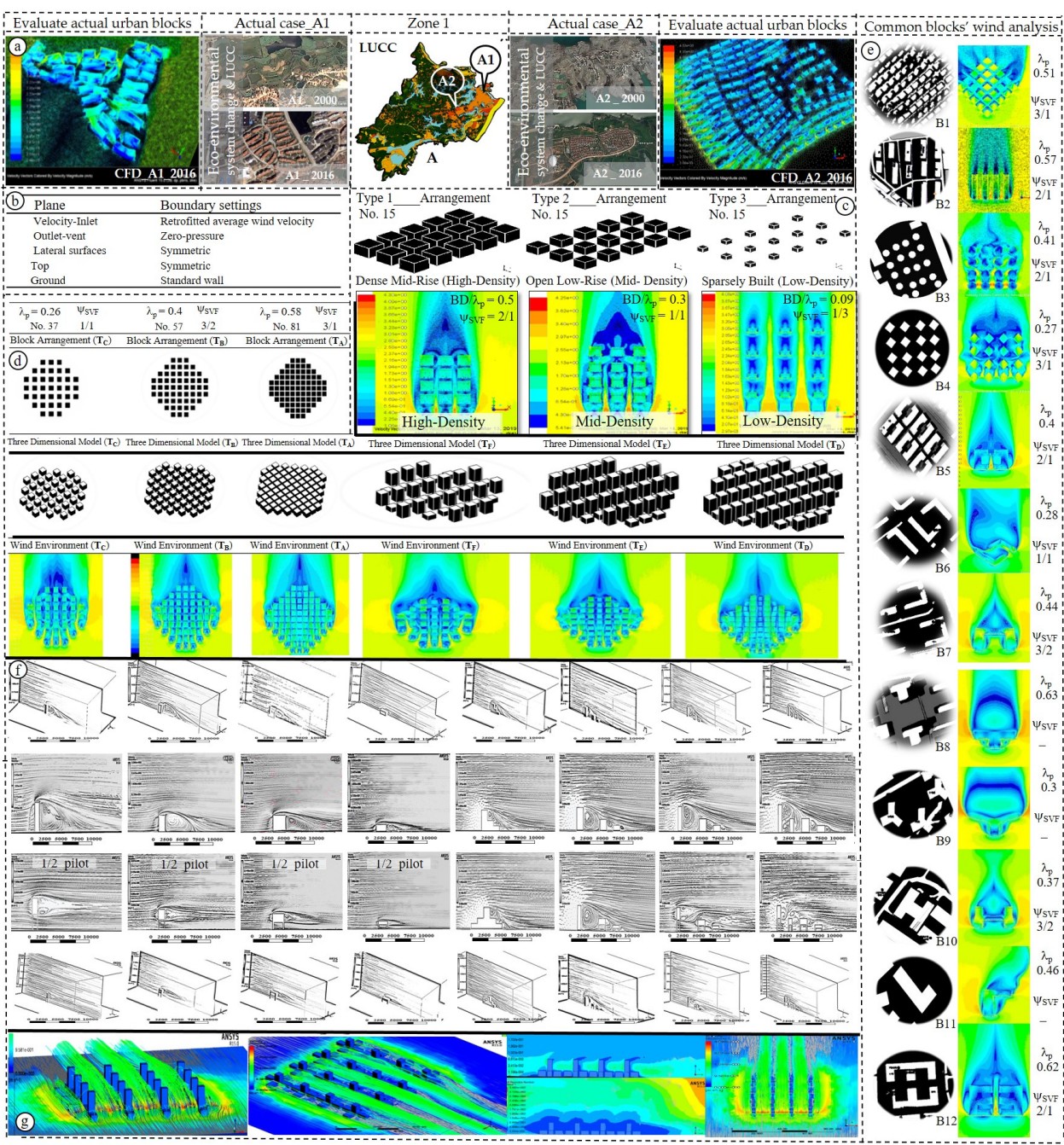

**Figure 7.** Block optimization solution to reinforce wind around the buildings and mitigate adverse effects of urbanization such as traffic pollutants, urban heat, UHII, and discomfort. (**a**) Analyzed and simulated actual A1 and A2 cases under extreme LUCC during urbanization. (**b**) Boundary conditions utilized in the simulation study around the building. (**c**) Modeled high-density, mid-density, and low-density blocks using diverse $\lambda_P$ and $\psi_{SVF}$. (**d**) Additional block arrangements implementing $\lambda_P$, $\psi_{SVF}$, and $\mu BH$. (**e**) Wind analysis around the actual buildings extracted from morphological study. (**f**) Further analysis of morphological and building effects on wind environments. (**g**) Optimized block.

### 3.2.2.2. Modeling Based on $\psi_{SVF}$ and $\lambda_P$

Due to the high spatial variability in built-up areas, major forms of dominant urban block configurations are examined based on $\psi_{SVF}$ and $\lambda_P$ (Figure 7e). As the air movement can control the ventilation manner around the urban blocks, the air movement patterns were investigated within elected cases which are residential blocks. Cuboid-like models

with low-density ($\lambda_p$ = 0.09), mid-density ($\lambda_p$ = 0.3), and high-density ($\lambda_p$ = 0.5) among the most common patterns were also analyzed and the results are outlined in Figure 7c. These models represent sparsely built-up areas with $\lambda_p$ = 0.09 and $\psi_{SVF}$ = 1/3, open low-rise with $\lambda_p$ = 0.3 and $\psi_{SVF}$ = 1/1, and dense mid-rise with $\lambda_p$ = 0.5 and $\psi_{SVF}$ = 2/1. Outcomes indicate that an airflow channel with a proper velocity is achievable within sparsely built areas with a lower density than in other cases. However, wind speed is lessened with the increase in recirculation in wakes behind obstacles. In open low-rise, when the $\psi_{SVF}$ is 1/1 and the obstacles become closer, the pressure difference becomes higher compared with the model with, $\lambda_p$ = 0.09 and $\psi_{SVF}$ = 1/3, which accelerates velocity between the blocks. The circulation of wakes behind buildings is moving faster but the issue is that when the static pressure becomes lower at the channel, the turbulence becomes higher at the end. In dense mid-rise where $\psi_{SVF}$ is 2/1, the static pressure at the first row of buildings is higher and because of the aspect ratio, incoming wind velocity declined. It is observed that turbulent kinetic energy and turbulent intensity increase as airflow moves across the blocks. In the densest arrangement, the tip area has a vertical axis according to the movement separation away from the front surface. This effect can help to lighten the heat and air pollutants occupied in the street canyon layer which can improve by vertical relations and mixing. This manner is comparable to that of the vortex in the arrangement of $\lambda_p$ = 0.3, which implies that there is a skyward flow distributing pedestrian layer pollutants and heat to the top spaces. In all cases, the varying vortex was generated with a spiraling air flow in canyons, in which air is pushed upward. These can be considered to release heat and pollutants accumulated between surfaces. The patterns of the air movement show that adverse air, containing pollution, and high temperatures exacerbated by canyons is intensely linked to the BD/$\lambda_p$ within the block arrangement. Eddy forms at various areas of the arrangement pack (the first canyon, central canyon, and last canyon) revealed clear similarities that should be considered in block scale design. Thus, the results indicate that air movement patterns around building bunches are more determined by positional geometry than by total block morphology.

Figure 7d shows further block density analysis integrating with μBH (average building height of $BH^A$, $BH^B$, and $BH^C$). A variable plot ratio is considered to examine changes around the blocks and the average flow in the channel is investigated regarding various $\lambda_p$ equal to 0.26, 0.4, and 0.58. Blocks are formed by a different number of buildings. A total of 81, 57, and 37 buildings in regular and irregular models were examined in high-density, mid-density, and low-density blocks, respectively. Overall, results determine that both horizontal and vertical speeds decline along with the wind flow direction which notes that urban blocks perform an approximately considerable resistance strength on the air movement. $\lambda_p$, $\psi_{SVF}$, and μBH parameters have a substantial impact on wind speed which, by changing these key factors, can control the air movement around the buildings and also can make an effective wind tunnel induced by obstacles. As can be seen in Figure 7d, a better wind environment arose when $\psi_{SVF}$ is 3/1. $\lambda_p$ is about 0.58, which is higher than other cases with a larger obstacle of about 81. This is evidence of the importance of plot ratio in design which in high $\lambda_p$ can aid the air circulation within blocks. By adapting μBH, and altering $\psi_{SVF}$ to 3/1, 6/1, and 9/1 for low-rise, mid-rise, and high-rise blocks, respectively, wind corridors can be generated. This can increase ventilation and improves the wind environment within the urban block.

The case comparisons also showed that the average velocity remained relatively the same horizontally from row to row when the wind crossed above the mid-rows. It is observed that when $\lambda_p$ is moderately low, the average movement and turbulence areas inside the canopy obtain a close balance after three rows of building. By frontal area intensification, the movement of the air in the corridor intensively declines with the acceleration of drag force vertically. A detailed analysis shows that air movement deterioration within the channel was experienced by each type of block arrangement. At the end of the block arrangement, canopy resistance abruptly vanishes, which is caused by the downstream wind. In addition, an air package from lateral arrangements increases the vertical flow.

### 3.2.2.3. Typology and Building Form Analysis

As one of the key physical elements of a city, buildings are an important part of urban morphology, contributing to its formation [74]. Forms of the buildings may lead to creating pressure differences around buildings, thereby they can be used to optimize wind conditions in their surroundings. As shown in Figure 7e, a typological and building form analysis was conducted to obtain a better understanding of the structural effects on the wind environment. Hence, different typologies with the major building forms in Jiangxia and Caidian as the most remarkable areas of Wuhan were selected from the street map of Baidu (www.map.baidu.com) and assessed. The intention is to discover effective manners to optimize ventilation potential outdoors within urban block scales. Wake flow affected by the building form is also considered with a view of wind potential. All models are simulated with the same condition in diverse forms to discover the differences. The pattern of rear wind and the movement of the air is dissimilar for each case, and it can be explained that each form has its particular identity by the special effect on the wind environment.

Simulation results declared that rear wind in each case has a distinct pattern that can be considered in the design of urban blocks. Type number one (B1) is one of the domain typologies with a high-density area which, by changing the cube to the diamond form, can break the motionless rear wind at the end of the array. B1 has negative quality in the lateral rows (six and seven) but can be modified by the $\psi_{SVF}$ adjustment. B2, because of the lower cross-section in the face of the wind, has better circulation in the canyon despite it having a higher $\lambda_p$ than the B1 type (1/3 of the canyon length at the faced part of wind flow has experienced a higher wind velocity of approximately 1 m/s). The pattern of the rear wind in B2 is a linear shape and it is approximately 2/3 of the rectangle length.

B5 has a lower $\lambda_p$ (0.4) which has different openings at the beginning (2/1), middle (1/5), and end (1/2). It revealed that openings have the potential to accelerate wind velocity by creating wind channels. Crossed wind channels formed by two openings in the courtyard increased the wind flow by 0.4 m/s, which in turn, accelerated at the entrance by 0.6 m/s but the flow was motionless at the end. Form B6 with a lower $\lambda_p$ (0.28) was set to face the wind flow at the corner side generating more drag along the sides and improving fluid motion behind the structure.

In the H shape (B10), static pressure at the inlet increased and dynamic pressure accelerated at the sides. Moreover, turbulent dissipation rate $\varepsilon$ appeared at the sides that faced the wind direction. Turbulence kinetic energy arose around the shape with more stress at the inlet sides. Accordingly, the ventilation rate largely declined at the windward side and raised behind it. In the L shape (B11), the round vortex did not appear but the pattern of a motionless wind was generated and performs with three main single linear wakes in which dynamic pressure is less than turbulent intensity. In the Y shape (B9), the boundary pattern of a thin circle of weak air was generated at the backside of the geometry, the circulation was much better at the central part where relative total pressure is at the average level. This air circulation pattern in the T shape (B8) faced a thick circle of weak air with a lower proper circulation at the backward end of the geometry because the relative total pressure fell. In the E shape (B12), the turbulent dissipation rate $\varepsilon$ and turbulent kinetic energy (TKE) in the edges of geometry and the opening towards the wind increased. Despite winding channels being formed across openings, air circulation within grooves is unsatisfactory and flow velocity declined because of turbulent intensity and dynamic pressure.

Figure 7c illustrates how generated vortexes were compared in different heights and openings to improve air circulation understanding. Different building heights such as high-rise (BH$^A$), mid-rise (BH$^B$), and low-rise (BH$^C$) were chosen individually and along each other in series with a different order. This is a comparison of the effect of the blocks over the ambient airflow which shows that the ambient air can be affected not just by the arrangement of street structures but also by the formation of ambient structures. Firstly, four styles of isolated buildings in different categories were studied, and afterward, they were investigated by using pilots. In the case of BH$^A$, which is taller than others, air

circulation is higher and the wake is longer than in other cases, and has a larger TKE at the top head of the structure. This form can reinforce the airflow downward and the created wake flow is over three times the height. $BH^B$ style just enhances the wind over two times and the lowest drag is when the height is $\frac{1}{2}$ and the width of the obstacle is one. In the last case (a combination of the three), a tall building on the left side accelerates the leeward structure of the upstream canyon. By adding the pilot in the style, altered vortexes around the structure aid to improve the air movement in the simulation area. The analysis of the above-mentioned wind circulation mechanism provides evidence that improving natural ventilation in dense areas can be achieved by using a new adaptable form within urban blocks. As a result, the optimized block is proposed in Figure 7g. This resulted in an effective air movement enhancement and heat reduction via improved cooling load between other urban building configurations. Further, the reduction in outdoor air temperature/Ta and solar gain by implementing the building's height diversity with μBH resulted in improved wind performance and a controlling cooling load of the suggested model. This not only regulates the radiation pattern but also thermal comfort and heat balance through airflow distribution.

## 4. Conclusions

Urbanization in China increased in speed following the initiation of the reform and opening policy. As a result of rapid development and the LUCC, Wuhan's urban morphology has changed substantially since the 1980s. Increased conversion of natural lands and water surfaces into urban land resulted in a reduction in natural ventilation, urban heat increase, and air pollution problems, particularly in rapidly growing urban fringe areas. This study investigated (1) the long-term effect of rapid urbanization and LUCC on urban morphology, climate, and the environment from 1980 to 2016; (2) Wuhan's built-up block configuration and building morphology focused on its urban fringe as growth-oriented areas; (3) and the optimizing block spatial configurations through an eco-sustainable design strategy to improve environmental performance towards climate change adaptation. The study aims to mitigate urban heat and LUCC's negative effects in rapidly urbanized areas. The findings are as follows:

- The urban morphology of Wuhan is highly affected by its rapid development, with significant changes in developing areas of Jiangxia, Caidian, Hunagpi, and Xinzhou. In 1990, LUCC was accompanied by a dramatic urban land increase (489.3 km$^2$) and the water surface reduction (145.64 km$^2$) became more severe in the urban fringe. Climate changes in the Jiangxia region were most pronounced with higher RH (4.05%) and Ta (0.7%). Long-term meteorological data analysis of urban fringe revealed that Ta increased by 0.4 °C on average per decade. This increases the severity of UHII in urban fringes.
- Different building types of $BH^A$, $BH^B$, and $BH^C$ indicate how common building types/morphologies and block configurations affect Ta and RH's differences over time. Integrating the performance characteristics of these influential urban design elements can effectively be used to control and optimize outdoor ventilation and thermal performance. Hence, normalizing $\lambda_p$, $\psi_{SVF}$, and μBH was found crucial for improving urban microclimates in new development projects which can enhance air circulation and thus mitigate heat around buildings.
- The simulated models' patterns of the air movement show that adverse air, containing pollution and high temperatures exacerbated by canyons, is intensely linked to the $BD/\lambda_p$ within the block arrangement. Eddy forms at various areas of the arrangement pack (the first canyon, central canyon, and last canyon) revealed clear similarities that require a combination with μBH (average building height of $BH^A$, $BH^B$, $BH^C$) for optimizing block scale design. The findings indicate that air movement patterns around building bunches are more determined by positional geometry than by total block morphology, allowing for climate change adaptation.

According to the air circulation pattern differences in each building with diverse typologies as an important part of urban morphology, further analysis is required in the combined three-in-one regulation strategy discussed in this study based on $BH^A$, $BH^B$, and $BH^C$. Advancement of the proposed eco-sustainable design strategy can be highly considered towards climate change adaptation to improve environmental performance under rapid urbanization, which is highly relevant to designers and policymakers.

**Author Contributions:** Conceptualization, M.M. and P.F.Y.; methodology, M.M. and Z.K.; software, M.M.; validation, M.M., X.O. and J.F.; formal analysis, M.M.; investigation, M.M., Z.K., E.d.l.J.H., W.L. and H.C.; resources, M.M. and P.F.Y.; data curation, M.M.; writing—original draft preparation, M.M.; writing—review and editing, M.M.; visualization, M.M.; supervision, P.F.Y.; project administration, M.M. and P.F.Y.; funding acquisition, M.M. All authors have read and agreed to the published version of the manuscript.

**Funding:** This article is supported by the Shanghai Science and Technology Committee (Grant No. 21DZ1204500), National Natural Science Foundation of China (Grant No. U1913603), the Shanghai Municipal Science and Technology Major Project (2021SHZDZX0100), and the Fundamental Research Funds for the Central Universities.

**Institutional Review Board Statement:** Not applicable.

**Informed Consent Statement:** Not applicable.

**Data Availability Statement:** Not applicable.

**Conflicts of Interest:** The authors declare no conflict of interest.

## Appendix A

**Table A1.** The land-use classification of Wuhan.

| Land Use Classification | | |
| --- | --- | --- |
| **No** | **Category** | **Explanation** |
| 1 | Farmlands/Arable land (1) | Mentions the cultivation of crops in the land, including cultivated land, new land reclamation, leisure, intermittent, grassland crop; to grow crops mainly farmer, agricultural mulberry, agricultural and forestry land; farming more than three years of beach and tideland. |
| 2 | Paddy Fields (11) | Mentions water conservation and irrigation equipment, in the common year to standard irrigation, for rice farming, lotus root and other aquatic crops such as arable land, including the implementation of rice and dryland crop rotation of cultivated land. |
| 3 | Dry-land (12) | Mentions irrigated water and facilities, by natural water crops to grow crops; water and watering facilities, in the general year under the normal irrigation of dry crop arable land; to cultivate the main cultivated land; the normal rotation of the leisure and Intermittently. |
| 4 | Forest/woodlands (2) | Mentions the growth of trees, shrubs, bamboo, coastal mangrove land and other forestry lands, as well as Refers to cannabis> 30% of natural and planted forests. Including timber forests, economic forests, shelter forests, and other forest lands. |
| 5 | Shrubbery lands (22) | Mentions the canopy height> 40%, the height of 2 meters below the dwarf forest and shrubland. |
| 6 | Sparse woodland (23) and Other woodlands (24) | 23)Mentions the forest canopy closure of 10–30% of the woodland.24) Refers to not forest afforestation, trails, nursery and all kinds of the garden (orchard, mulberry, tea, hot forest garden, etc. |
| 7 | Grassland (3) | Mentions the growth of herbaceous plants, covering more than 5% of the various types of grassland, including the main grassland and canopy density10% of the sparse grassland. |

Table A1. *Cont.*

| | Land Use Classification | |
|---|---|---|
| **No** | **Category** | **Explanation** |
| 8 | High-coverage grassland (31) | Mentions > 50% of the natural grass, improved grass and lawn. Such grassland water conditions are generally good, and the grass is growing dense. |
| 9 | Covered grassland (32) | Mentions the coverage in the 20–50% of the natural grass and improved grass, such grass is generally insufficient water, the grass is more sparse. |
| 10 | Low-coverage grassland (33) | Mentions the coverage of 5–20% natural grassland. Such grassland lacks water, and the grass is sparse, and poor use of animal husbandry conditions. |
| 11 | Waters (4) | Mentions natural land and water conservancy facilities. |
| 12 | River canal (41) | Mentions the natural formation or man-made cavity of the river and the backbone of the yearly water level under the land. Manufactured channels include embankments. |
| 13 | Lake (42) | Mentions the natural formation of the water area below the perennial water level. |
| 14 | Reservoir pond (43) | Mentions the structure of the water storage area below the twelve-monthly water level of the land. |
| 15 | Permanent glacier snow(44) | Mentions the sustained glaciers and snow coated by the land. |
| 16 | Beach (45–46) | 45)Mentions the coastal tide of high tide and low tide between the tidal zone.46) Refers to the river and lake waters and the flood level between the water level between the land. |
| 17 | Built-up and rural areas, industrial and mining, and residential land (5) | Mentions city and rural residential zones, industrial, mining, transportation, and other lands. |
| 18 | Built-up land (51) | Mentions large, medium and small cities and counties above the built space land. |
| 19 | Rural settlements (52) | Mentions rural areas other than towns. |
| 20 | Other structure lands (53) | Mentions factories and mines, large industrial areas, oil, salt, quarry and other land and traffic roads, airports and special land. |
| 21 | Unused land (6) | Land that has not yet been used, including difficult land. |
| 22 | Sandy land (61) | Mentions the surface for the sand cover, vegetation coverage below 5% of the land, together with the desert, not involving the desert in the water system. |
| 23 | Gobi (62) | The ground surface is dominated by crushed gravel, with vegetation covering under 5%. |
| 24 | Saline (63) | The land with saline and alkali-tolerant plants can only be grown when the surface is concentrated, and the vegetation is scarce. |
| 25 | Swamp (64) | Mentions horizontal low-lying, weak drainage, long-term wet, seasonal stagnant water or perennial water, surface growth of wet plants of the land. |
| 26 | Bare Land (65) | Mentions land covered by soil and vegetation coverage under 5%. |
| 27 | Bare Rock Texture (66) | Mentions the surface of rock or gravel, covering the area of >5% land. |

Source: Wuhan Land Resource and Planning Bureau (WLRPB).

## Appendix B

A digital model for each of the residential blocks was constructed and an ANSYS Fluent CFD simulation was conducted. A k-ε model was used and set as the standard model which was carefully chosen from the viscous model to set parameters for turbulent flow. The standard governing equations of incompressible turbulent airstream circulation around building models were connected and the RANS equations (motion for fluid flow used to explain turbulent flows) are described as follows:

$$\rho(u.\nabla)u = \nabla.[-\rho l + (\mu + \mu_T)(\nabla u + (\nabla u)^T) - \frac{2}{3}(\mu + \mu_T)(\nabla.u)l - \frac{2}{3}\rho k l] + f \quad \text{(A1)}$$

$$\nabla.(\rho u) = 0 \tag{A2}$$

$$\rho(u.\nabla)k = \nabla.[(\mu + \frac{\mu_T}{\sigma_k})\nabla k] + P_k - \rho\varepsilon \tag{A3}$$

$$\rho(u.\nabla)\varepsilon = \nabla.[(\mu + \frac{\mu_T}{\sigma_\varepsilon})\nabla\varepsilon] + C_{e1}\frac{\varepsilon}{k}P_k - C_{e2}\rho\frac{\varepsilon^2}{k} \tag{A4}$$

$$\mu_T = \rho c_\mu \frac{k^2}{\varepsilon} \tag{A5}$$

$$\rho_k = \mu_T[\nabla u : (\nabla u + (\nabla u)^T) - \frac{2}{3}(\nabla u)^2] - \frac{2}{3}\rho k\nabla.u \tag{A6}$$

where $\mu$ is the fluid's dynamic viscosity (kg/ms); $\mu_T$ is the turbulent viscosity (kg/ms); $u$ is the velocity field (m/s); $\rho$ is the density of the fluid (kg/m$^3$); $P$ is the pressure (Pa); $k$ is the turbulent kinetic energy (m$^2$/s$^2$); and $\varepsilon$ is the turbulent dissipation rate (m$^2$/s$^3$) of the turbulence model. The model constants in this simulation method are shown in Table A2.

**Table A2.** Model constants derived from experimental data used in the governing equations.

| Constant | Value |
| --- | --- |
| C$\mu$ | 0.09 |
| C$e_1$ | 1.44 |
| C$e_2$ | 1.92 |
| $\sigma_k$ | 1.0 |
| $\sigma_\varepsilon$ | 1.3 |

These approaches are used to reflect the differences in the influence of different types of block morphology (block form; HB$^A$, HB$^B$, and HB$^C$) on the urban microclimate and wind environment which can provide targeted strategies for urban design to improve and regulate urban microclimates.

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
