# Peer review of "Urban Heat Mitigation towards Climate Change Adaptation: An Eco-Sustainable Design Strategy to Improve Environmental Performance under Rapid Urbanization"

_atmosphere, doi:10.3390/atmos14040638_

Round 1

Reviewer 1 Report

Urban heat mitigation towards climate change adaptation: An eco-sustainable design strategy to improve environmental performance under rapid urbanization

 In this study, the authors conduct a (1) long-term LUCC and metrological analys in Wuhan metropolitan area from 1980 to 2016, (2) the role and relevance of optimizing building morphology and urban block configuration were discussed to mitigate the adverse effects of LUCC under speedy development process, (3) particular design attention in strategy towards climate change adaptation for environmental performance improvement was paid in Wuhan’s fast-growing zones of Jiangxia, Caidian, Hunagpi, and Xinzhou. The results show that UHII in 1980 was less severe than in 2016. It is found obligatory for nature-based design to adopt urban morphology indicators (UMIs) such as average building height (μBH), sky view factors (SVF), and building density (BD/p=% of built area) towards these changes. As a result of this study's strategy, urban heat is mitigated via reinforcing wind in order to adapt to climate change, which impacts the quality of life directly in developing areas.

The topic presented is interesting, and the paper is quite well structured to a relevant audience. I can recommend to publication once the following concerns are properly addressed:

·         Do you introduce the acronym CFD in the abstract without explaining what is into advance?.

·         In the abstract, the authors argue “long-term LUCC and metrological analyses” -> Do you mean “meterorological” or just “quantitative”?

·         When the authors argue their aim in the abstract “In this study, (1) long-term LUCC and metrological analyses were conducted in Wuhan metropolitan area from 1980 to 2016, (2) the role and relevance of optimizing building morphology and urban block configuration were discussed to mitigate the adverse effects of LUCC under speedy development process, (3) particular design attention in strategy towards climate change adaptation for environmental performance improvement was paid in Wuhan’s fast-growing zones of Jiangxia, Caidian, Hunagpi, and Xinzhou”. This is very broad sentence. The authors must be much sharper and summarize the aim much better and in a clear way.

·         The introduce starts in this way “Urban heat island (UHI) effects were first noticed in London in the 19th century [1]. UHI effect refers to the phenomenon that urban air temperature is considerably higher than its surroundings and rural areas due to the thermal properties of the urban tissue [2].” I am a little bit skeptical to introduce this in this way. I would recommend you to do in the opposite way, replying in this way -> What is UHI and why does it a societal concern? This is my suggestion to develop this part: “Scale economies require more agglomeration of people and goods -> The world is increasingly urbanized -> Rapid urbanization without the right planification -> Undesired effects such heat islands -> These generate a lot of problems and health issues ”

·         When the authors refer to “rapid urbanization”, they should refer to concepts more accepted in literature such “pseudo-urbanization” or “fast urbanization”, referring to those urbanization processes that not regulated. (These concepts are introduced in the paper entitled Mapping Population Dynamics at Local Scales Using Spatial Networks)

·         The authors “As one of the most significant adverse effects of rapid urbanization [3], UHI arising from land use/cover changes (LUCC) [4] has been reported to cause many alarming problems closely related to humans and the environment: deteriorating air quality [5], threatening human comfort and well-being [6], increasing energy and water consumption [7, 8], etc. “ The authors refer to LUCC such the cause of alarming problems, and this is not always true. For example, the forest area in many developed countries is higher than ever. It is also a trend to observe how many cities in wealthy countries are removing the car out and broadening their green spaces, for example. This happens in many European and US cities.

·         The authors argue: “According to the U.N. 2018 World Urbanization Trend (https://population.un.org/wup/DataQuery/), global urbanization rose from 24% in 1950 to 55% in 2018. As an inevitable process of human development, this trend is expected to reach 68%...” Under my perspective is not right to include a reference in that way. Instead, you should refer to (UN Forecast, YEAR).

·         The authors argue: “Apart from these naturally derived UHII mitigation and control measures, urban geometry [24], building density (BD/lp=% of built area) [25], aspect ratio (building height/street width, H/W) [26], built environment albedo increment (e.g. reflective pavement) [1], layout, form and fabric [27-29] would also affect local climate and thermal comfort.” This is a very interesting insight from this paper. I agree. I would recommend also relating these metrics to streets (relationship built heights with street width). This kind of metrics are crucial  for estimating UHI. I recommend you to check and include in your list the study “Using street based metrics to characterize urban typologies”, but also other similar in this aspect.

·         They also argue “Studies indicated that the ventilation performance of blocks is greatly influenced by their configuration [30, 31], which further affects the dispersion of air pollutants and UHII.” What is the importance of the materials used in the construction of buildings? Could you check this with remote sensing?

·         In order to link urban spatial configuration to UHII, different urban morphology indicators (UMIs), such as average building height (μBH), sky view factors (ySVF), etc., have been developed and used [32, 33]” and after that “Therefore, remote sensing studies cannot establish a direct relationship between human thermal comfort and urban morphology.”-> How do the authors can extract 3D urban morphology using remote sensing? I would suggest they should use LiDAR data or the combination with aerial imagery/remote images. I recommend you the study “Land-use mapping of Valencia city area from aerial images and LiDAR data” that can be an optimal guidance for analyzing how LUCC can be tracked with aerial imagery, remote sensing and laser scanning data that allows getting the 3D information (such building height that is a fundamental factor for UHI). The authors should deep this point in the work.

·         Following the relationship outlined above, corresponding guidelines and recommendations for UHII mitigation were presented to decision makers for developing cities that are climate-adaptive. Particularly, many related studies in the field of remote sensing have been conducted by exploring the relationship between LST and vegetation coverage ratio/spatial pattern, fabricating 2D/3D morphology characteristics [34-36]. However, a major flaw lies in the fact that LST obtained from remote sensing is not a direct parameter signifying thermal sensation. Therefore, remote sensing studies cannot establish a direct relationship between human thermal comfort and urban morphology. Besides, air quality is linked to human comfort [37], and a low quality may result in lower pollutant distributions and higher UHIIs in actual urban blocks, which makes their relationship to block configurations crucial. Further research is still needed to clarify and grasp this relationship due to complicated morphologies in complex urban environments.

·         Several studies combined measurements on-site with information collected via remote sensing [38, 39]: climatic parameter sensors were placed in the study area, and remote sensing technology collected morphological data surrounding the measuring points.

·         However, this method's reliability and precision depend on where the measuring points are placed. A study area's thermal conditions may not be accurately represented by meteorological parameters such as Ta obtained from one or more measurement sites. Luckily, advancements in computer technology and numerical calculation have made it possible to conduct reliable studies and simulation experiments in order to understand more thoroughly the correlation between urban morphology and local thermal comfort. In addition to providing detailed information on the thermal comfort of the study area, the computer simulation model can also assess the relationship between urban morphology and thermal comfort in built-up areas [40, 41]. In urban planning and design, the most influential urban form and layout can also be explored [42, 43].

·         The case study presented in focused on the city of Wuhan. The authors argue at some point: “Urbanization in China increased in speed following the initiation of the reform and opening policy. As a result of rapid development and the LUCC, Wuhan's urban morphology has changed substantially since the 1980s. Increased conversion of natural lands and water surfaces into urban land resulted in a reduction of natural ventilation, urban heat increase, and air pollution problems, particularly in rapidly growing urban fringe areas.”. I would recommend to be more careful with this adding a more clear contextualization of this particular study area answering questions such how much fast urbanization is there and what is the context of urbanization in China during last years. Pentland et al. (2019) analyzed precisely this issue (Globalization and the shifting centers of gravity of world's human dynamics: Implications for sustainability) showing how urbanization processes at a global scale were mostly focused in China during the last years. The particular context of China must be introduced showing why the case study selected is relevant for an international perspective.

·         When the authors introduce the study area “The city of Wuhan, with latitude 30°N–30.7°N and longitude 114°E–114.55°E (built-up area),” they should not include coordinates. Instead they should introduce the relative location (inner China, for example)

·         How the results presented here can be extracted to other cities?

·         About the graphical part, some of the figures are not easily readable. In the maps, I am missing a proper cartographic treatment including spatial scales, for example.

Author Response

Response to Reviewer 1 Comments

Point 1: Do you introduce the acronym CFD in the abstract without explaining what is into advance?

Response 1: Thank you so much for your comment. There is a discussion of CFD (Computational Fluid Dynamics) on page 3, line 111, which is widely used in related studies. CFD is now commonly mentioned in the abstracts, just like GIS, in today's research. Therefore, we stated "integrating GIS-CFD…" in the abstract. There are many examples such as:

  1. CFD modelling of hydrogen and hydrogen-methane explosions – Analysis of varying concentration and reduced oxygen atmospheres” with the following DOI: https://doi.org/10.1016/j.jlp.2023.105012
  2. Parametric aeroelastic modeling, maneuver loads analysis using CFD methods and structural design of a fighter aircraft” with the following DOI:

https://doi.org/10.1016/j.ast.2023.108231

  1. CFD modeling of the building integrated with a novel design of a one-sided wind-catcher with water spray: Focus on thermal comfort” with the following DOI:

https://doi.org/10.1016/j.seta.2022.102736

  1.  

Point 2: In the abstract, the authors argue “long-term LUCC and metrological analyses” -> Do you mean “meterorological” or just “quantitative”?

Response 2: Thank you very much for your insightful comment. A few letters were missed, and the correction has been made. Therefore, "meteorological" is correct and we have rewritten it in response to your valuable insight. Now, it can read as follows “long-term LUCC and meteorological analyses”. Please check it on page 1, line 18.

Point 3: When the authors argue their aim in the abstract “In this study, (1) long-term LUCC and metrological analyses were conducted in Wuhan metropolitan area from 1980 to 2016, (2) the role and relevance of optimizing building morphology and urban block configuration were discussed to mitigate the adverse effects of LUCC under speedy development process, (3) particular design attention in strategy towards climate change adaptation for environmental performance improvement was paid in Wuhan’s fast-growing zones of Jiangxia, Caidian, Hunagpi, and Xinzhou”. This is very broad sentence. The authors must be much sharper and summarize the aim much better and in a clear way.

Response 3: Thank you very much for your valuable comment. The following revisions were made to make the study's aim more clear:

 “In this study, (1) long-term LUCC and metrological analyses were conducted in Wuhan metropolitan area from 1980 to 2016, (2) to mitigate the adverse effects of LUCC under speedy development process, the role and relevance of optimizing building morphology and urban block configuration were discussed, (3) particular design attention in strategy towards climate change adaptation for environ-mental performance improvement was paid in Wuhan’s fast-growing zones.

Please check it on page 1, lines 18-24.

Point 4:  The introduce starts in this way “Urban heat island (UHI) effects were first noticed in London in the 19th century [1]. UHI effect refers to the phenomenon that urban air temperature is considerably higher than its surroundings and rural areas due to the thermal properties of the urban tissue [2].” I am a little bit skeptical to introduce this in this way. I would recommend you to do in the opposite way, replying in this way -> What is UHI and why does it a societal concern? This is my suggestion to develop this part: “Scale economies require more agglomeration of people and goods -> The world is increasingly urbanized -> Rapid urbanization without the right planification -> Undesired effects such heat islands -> These generate a lot of problems and health issues ”

Response 4: Thank you very much for your valuable comment. Accordingly, to stress your comment of “What is UHI and why does it a societal concern?”, we revised it as follows (lines 38-47):

Urban heat islands (UHI) are mostly caused by urbanization and climate change [1], which refer to the phenomenon that urban air temperature due to the thermal properties of the urban tissue is considerably higher than its surroundings and rural areas [2]. It has been found that urbanization processes at a global scale have largely been focused on China in recent years [3]. With an increase of rapid urbanization without the right planificationand (false or pseudo-urbanization processes [4]) and accelerating the process of land use/cover changes (LUCC), the UHI phenomenon has become increasingly more serious [5]. Undesired effects of UHIs generate many grave problems closely related to humans and the environment: deteriorating air quality [6], threatening human comfort and well-being [7], increasing energy and water consump-tion [8, 9], etc.

The first part is to reply to what is UHI, and the second shows why China is considered and how rapid urbanization without the right planification can make UHI more serious. Next, we emphasized its societal concern. The first sentence has also been updated with reference 1. Also, other references have been added such as references 3 & 4.

[1] Battista, G., et al., Effects of urban heat island mitigation strategies in an urban square: A numerical modelling and experimental investigation. Energy and Buildings, 2023. 282: p. 112809.

 Point 5:  When the authors refer to “rapid urbanization”, they should refer to concepts more accepted in literature such “pseudo-urbanization” or “fast urbanization”, referring to those urbanization processes that not regulated. (These concepts are introduced in the paper entitled Mapping Population Dynamics at Local Scales Using Spatial Networks)

Lines 40-43 has

    Response 5: Thank you very much for your valuable comment. We have added your abovementioned concepts of “pseudo-urbanization” or “false urbanization” according to the paper entitled “Mapping Population Dynamics at Local Scales Using Spatial Networks”. Accordingly, the changes have been made on page 1, lines 42-45, and reference number 4 is added as follows:

      “With an increase of rapid urbanization without the right planificationand (false or pseudo-urbanization processes [4]) and accelerating the process of land use/cover changes (LUCC), the UHI phenomenon has become increasingly more serious [5].”

    [4] Balsa-Barreiro, J., A.J. Morales, and R.C. Lois-González, Mapping Population Dynamics at Local Scales Using Spatial Networks. Complexity, 2021. 2021: p. 8632086.

       Point 6:  The authors “As one of the most significant adverse effects of rapid urbanization [3], UHI arising from land use/cover changes (LUCC) [4] has been reported to cause many alarming problems closely related to humans and the environment: deteriorating air quality [5], threatening human comfort and well-being [6], increasing energy and water consumption [7, 8], etc. “ The authors refer to LUCC such the cause of alarming problems, and this is not always true. For example, the forest area in many developed countries is higher than ever. It is also a trend to observe how many cities in wealthy countries are removing the car out and broadening their green spaces, for example. This happens in many European and US cities.

    Response 6: Thank you very much for your comment. It is important to note that rapid urbanization and urban land cover increment contribute to LUCC, and this study focused on these transformations. However, to follow your valuable comment, we have revised the text as follows: “With an increase of rapid urbanization without the right planificationand (false or pseudo-urbanization processes [4]) and accelerating the process of land use/cover changes (LUCC), the UHI phenomenon has become increasingly more serious [5]. Un-desired effects of UHIs generate many grave problems closely related to humans and the environment: deteriorating air quality [6], threatening human comfort and well-being [7], increasing energy and water consumption [8, 9], etc.” Please check page 1, line 43.

    This includes your valuable comments as well such as “(false or pseudo-urbanization processes [4])”.

    Point 7:  The authors argue: “According to the U.N. 2018 World Urbanization Trend (https://population.un.org/wup/DataQuery/), global urbanization rose from 24% in 1950 to 55% in 2018. As an inevitable process of human development, this trend is expected to reach 68%...” Under my perspective is not right to include a reference in that way. Instead, you should refer to (UN Forecast, YEAR).

    Response 7: Thank you very much for your valuable comment. To follow your comment, we have revised the text to reflect your remarks: “According to the United Nations' Global Urbanization Trend (UN Forecast, 2018) report [10], global urbanization rose from 24% in 1950 to 55% in 2018. As an inevitable process of human development, this trend is expected to reach 68%...”. Please check lines 47-49.

      Point 8:  The authors argue: “Apart from these naturally derived UHII mitigation and control measures, urban geometry [24], building density (BD/lp=% of built area) [25], aspect ratio (building height/street width, H/W) [26], built environment albedo increment (e.g. reflective pavement) [1], layout, form and fabric [27-29] would also affect local climate and thermal comfort.” This is a very interesting insight from this paper. I agree. I would recommend also relating these metrics to streets (relationship built heights with street width). This kind of metrics are crucial  for estimating UHI. I recommend you to check and include in your list the study “Using street based metrics to characterize urban typologies”, but also other similar in this aspect. The street metrics (relationship built heights with street width) can generate urban typologies is highly important for estimating UHI.

    Response 8: Thank you very much for your valuable comment. According to the comment, the following was stated, and the citation has been added: “Meanwhile, the relationship between built heights and street widths (street metrics) is highly important to estimate UHI and formulate urban typologies [33].”

    [33] Hermosilla, T., et al., Using street based metrics to characterize urban typologies. Computers, Environment and Urban Systems, 2014. 44: p. 68-79.

    Point 9: They also argue “Studies indicated that the ventilation performance of blocks is greatly influenced by their configuration [30, 31], which further affects the dispersion of air pollutants and UHII.” What is the importance of the materials used in the construction of buildings? Could you check this with remote sensing?

    Response 9: Thank you very much for your comment. However, the materials used in the construction of buildings are not a part of the study or its focus, which may be explored in a future study. Adding this part may change the focus of the current study.

    Point 10: In order to link urban spatial configuration to UHII, different urban morphology indicators (UMIs), such as average building height (μBH), sky view factors (ySVF), etc., have been developed and used [32, 33]” and after that “Therefore, remote sensing studies cannot establish a direct relationship between human thermal comfort and urban morphology.”-> How do the authors can extract 3D urban morphology using remote sensing? I would suggest they should use LiDAR data or the combination with aerial imagery/remote images. I recommend you the study “Land-use mapping of Valencia city area from aerial images and LiDAR data” that can be an optimal guidance for analyzing how LUCC can be tracked with aerial imagery, remote sensing and laser scanning data that allows getting the 3D information (such building height that is a fundamental factor for UHI). The authors should keep this point in the work.

  1. Following the relationship outlined above, corresponding guidelines and recommendations for UHII mitigation were presented to decision makers for developing cities that are climate-adaptive. Particularly, many related studies in the field of remote sensing have been conducted by exploring the relationship between LST and vegetation coverage ratio/spatial pattern, fabricating 2D/3D morphology characteristics [34-36]. However, a major flaw lies in the fact that LST obtained from remote sensing is not a direct parameter signifying thermal sensation. Therefore, remote sensing studies cannot establish a direct relationship between human thermal comfort and urban morphology. Besides, air quality is linked to human comfort [37], and a low quality may result in lower pollutant distributions and higher UHIIs in actual urban blocks, which makes their relationship to block configurations crucial. Further research is still needed to clarify and grasp this relationship due to complicated morphologies in complex urban environments.
  2. Several studies combined measurements on-site with information collected via remote sensing [38, 39]: climatic parameter sensors were placed in the study area, and remote sensing technology collected morphological data surrounding the measuring points.
  3. parameters such as Ta obtained from one or more measurement sites. Luckily, advancements in computer technology and numerical calculation have made it possible to conduct reliable studies and simulation experiments in order to understand more thoroughly the correlation between urban morphology and local thermal comfort. In addition to providing detailed information on the thermal comfort of the study area, the computer simulation model can also assess the relationship between urban morphology and thermal comfort in built-up areas [40, 41]. In urban planning and design, the most influential urban form and layout can also be explored [42, 43].
  4. However, this method's reliability and precision depend on where the measuring points are placed. A study area's thermal conditions may not be accurately represented by meteorological

Response 10: Thank you very much for your comment. However, it is a bit confusing as these four subsections are from the manuscript. Regarding your main question “How do the authors can extract 3D urban morphology using remote sensing?”, if we understood you correctly, it's worth noting that here we reviewed other studies and the point was on the relationships, not the methodology for extracting 3D urban morphology using LiDAR and aerial imagery/remote images. However, we followed your comment to rewrite some parts and also add your reference. Therefore, the following revision has been made on page 2, lines 76-83: “ Particularly, many related studies in the field of remote sensing have been conducted by exploring the relationship between LST and vegetation coverage ratio/spatial pat-tern, fabricating 2D/3D morphology characteristics [38-40]. It can be combined with aerial images and LiDAR to track LUCC and obtain 3D information (such as building height, which is a fundamental factor in UHI) [41]. However, a major flaw lies in the fact that LST obtained from remote sensing is not a direct parameter signifying thermal sensation. Therefore, remote sensing studies cannot establish a direct relationship between human thermal comfort and urban morphology.

   Point 11:   The case study presented in focused on the city of Wuhan. The authors argue at some point: “Urbanization in China increased in speed following the initiation of the reform and opening policy. As a result of rapid development and the LUCC, Wuhan's urban morphology has changed substantially since the 1980s. Increased conversion of natural lands and water surfaces into urban land resulted in a reduction of natural ventilation, urban heat increase, and air pollution problems, particularly in rapidly growing urban fringe areas.”. I would recommend to be more careful with this adding a more clear contextualization of this particular study area answering questions such how much fast urbanization is there and what is the context of urbanization in China during last years. Pentland et al. (2019) analyzed precisely this issue (Globalization and the shifting centers of gravity of world's human dynamics: Implications for sustainability) showing how urbanization processes at a global scale were mostly focused in China during the last years. The particular context of China must be introduced showing why the case study selected is relevant for an international perspective.

    Response 11: Thank you very much for your valuable comment. We have cited (Globalization and the shifting centers of gravity of world's human dynamics: Implications for sustainability) and a very clear context added to show why China is selected according to its rapid urbanization. Therefore, the following revisions have been conducted on page 1, lines 40-42: “It has been found that urbanization processes at a global scale have largely been focused on China in recent years [3].

   [3] Balsa-Barreiro, J., et al., Globalization and the shifting centers of gravity of world's human dynamics: Implications for sustainability. Journal of Cleaner Production, 2019. 239: p. 117923.

   Point 12:    When the authors introduce the study area “The city of Wuhan, with latitude 30°N–30.7°N and longitude 114°E–114.55°E (built-up area),” they should not include coordinates. Instead they should introduce the relative location (inner China, for example)

    Response 12: Thank you very much for your valuable comment. The coordinate system with latitude 30°N–30.7°N and longitude 114°E–114.55°E shows the build-up area of the study. Thus, this can help the readers to understand more. Your comment, however, were taken into consideration and we revised the document accordingly. The revision can be seen on page 5, line 152 “The city of Wuhan (south central China)”.

  Point 13:  How the results presented here can be extracted to other cities?

    Response 13: Thank you very much for your comment. As we mentioned in the results and discussion section (3-1), to achieve urban design strategies, understanding chronological land use and climate change was essential for future sustainable urban design. Hence, a wide scale of land use transformation was analyzed under rapid urbanization. The method that we have used in this study to achieve urban design strategies is not just used in Wuhan but can be extended to study other developing areas. However, adaptation is crucial and the rest of the method is similar. A long-term study of LUCC using the land use classification of any city can be considered and not just Wuhan. Urban morphology is highly affected by rapid development, with significant changes in developing areas. This vision of the study also can be seen in all developing areas, which is not limited to Wuhan metropolitan areas. In general, this study combined methods from a large scale to a small scale, which can take place in all developing cities after adaptation. The strategy of using the relationship between building height and the configuration also can be considered widely, which we have already mentioned in the study. In lines 368-369 we have mentioned that BD/lp and ySVF consideration can affect the wind environment. This are really generalized and not just for Wuhan.

  Point 14: About the graphical part, some of the figures are not easily readable. In the maps, I am missing a proper cartographic treatment including spatial scales, for example.

    Response 14: Thank you very much for your valuable comment. To make them more readable, we have revised them as follows:

Spatial scales are added to the maps. Please check Figure 3 to see the changes.

Reviewer 2 Report

Dear Authors,

Thank you for the patience in this review process. The paper is of very good quality and the originality of the research design is very high. I am impressed at the level of complexity and the depth of the research carried out. Overall, I have no other comments than to check the paper for consistency and the formatting of the references. The figures also appear with small text inside, which should be increased to allow the reader to comprehend it. They are also quite complex, but I understand that they reflect the research carried out. In conclusion, I am satisfied with the research carried out and will recommend your paper for a moderate revision.

Kind regards,

Reviewer

Author Response

Dear Reviewer 2,

Thank you very much for your affirmative commnets regarding our study. Moderate revision has been done and due to no comments to answer, we would like to thank you again for your positive comments on this paper. 

Best regards,
